# Knowledge on Stroke Recognition and Management among Emergency Department Healthcare Professionals in the Republic of Cyprus

**DOI:** 10.3390/healthcare12010077

**Published:** 2023-12-29

**Authors:** Christos Rossis, Koralia A. Michail, Nicos Middleton, Maria Karanikola, Elizabeth Papathanassoglou, Meropi Mpouzika

**Affiliations:** 1Nicosia General Hospital Cyprus, 2031 Nicosia, Cyprus; c.rossis1998@gmail.com; 2Department of Nursing, Cyprus University of Technology, 3041 Limassol, Cyprus; koralia.michail@cut.ac.cy (K.A.M.); nicos.middleton@cut.ac.cy (N.M.); maria.karanikola@cut.ac.cy (M.K.); 3Faculty of Nursing, University of Alberta, Edmonton, AB T6G 1C9, Canada; papathan@ualberta.ca

**Keywords:** early arrival, emergency department, healthcare professionals, hospital care, knowledge, management, prehospital care, recognition, stroke

## Abstract

Stroke is a global leading cause of death and disability. Knowledge of related guidelines is crucial for emergency department (ED) staff, influencing early diagnosis and timely treatment. We investigated Greek Cypriot ED healthcare professionals’ (nurses and physicians) knowledge in recognizing and managing stroke. A descriptive cross-sectional study spanned November 2019 to April 2020, encompassing four private and seven public EDs in the Republic of Cyprus. The data were collected through a self-reported questionnaire developed by the research team, consisting of 37 questions. Eight questions focused on sociodemographic and employment characteristics, twenty-eight assessed knowledge in stroke recognition and management (each item was equally weighted without deliberate prioritization), and one question addressed self-assessment of knowledge in stroke care. A total of 255 nurses (response rate (RR): 74.1%) and 26 physicians (RR: 47.3%) completed the questionnaire. The average correct response rate was 12.9 out of 28 statements (SD: 4.2), with nurses and physicians scoring 12.6 (SD: 4.1) and 15.7 (SD: 4), respectively. Work experience significantly influenced stroke knowledge, with all groups demonstrating superiority over those with less than one year of experience. Participants with previous training scored an average of 1.45 additional correct answers while educational attainment did not significantly influence stroke knowledge. Investigating stroke knowledge among emergency department nurses and physicians in the Republic of Cyprus revealed significant deficits. This study stresses targeted interventions, including education, yearly examinations, workshops with hands-on training, and repeated training, to address these gaps and enhance the overall stroke care capabilities of the healthcare professionals.

## 1. Introduction

Stroke poses a significant healthcare burden due to morbidity, disability, and mortality [1,2,3]. Modern intraarterial therapies of ischemic stroke such as thrombolysis and thrombectomy have a narrow optimal therapeutic window, beyond which clinical outcomes deteriorate [4,5]. Therefore, timely recognition and management of patients suffering a stroke is vital, since it renders more patients as suitable candidates for such therapies.

Stroke detection education involves training healthcare professionals, spanning undergraduate, postgraduate, and ongoing professional development. Two programs, an online initiative by Angels’ Initiatives and the Advanced Stroke Life Support (ASLS) course, offer continual stroke education [6,7]. Guidelines on stroke diagnosis and management are regularly updated [5,8,9]. It is, however, a matter of ongoing research whether healthcare professionals (HCPs) involved in stroke care have up-to-date knowledge on this subject. Substantial research has been dedicated to prehospital stroke care, spanning from the assessment of knowledge [10,11] to the development of triage techniques [12] and targeted educational interventions for improving knowledge and optimizing stroke recognition. Another critical aspect is the fast transfer to dedicated stroke centers [13,14,15].

Equally pivotal is the timely admission of stroke patients to the emergency department (ED). Delays in first-contact diagnosis and interfacility transfer can be attributed to hospital-to-hospital transfer [16], while hospital door-to-revascularization delays are dependent mainly on ED staff, nurses, and physicians. These professionals represent the next set of healthcare professionals that the stroke patient encounters following paramedics [17,18]. Stroke patients might also present themselves straight to the ED, making ED personnel their first medical contact. Review articles have identified various barriers hindering the guideline-based management of stroke patients [19,20]. Among these barriers, insufficient knowledge and stroke unawareness are frequently encountered among personnel involved in stroke care, particularly in emergency departments (Eds), including both physicians [21,22,23,24,25,26,27,28] and nurses [22,23,24,25,29,30,31,32,33,34,35].

The phenomenon is universal; studies have explored ED personnel from diverse countries, including the U.S. [23,32,33,34,35], Great Britain [31], Scandinavia [21,25], India [30], Kenya [36], Saudi Arabia [27,28], and Australia [24,26]. These countries have different management approaches and various levels of stroke care. Most studies have quantified the level of knowledge regarding signs and symptoms of stroke and eligibility criteria for intraarterial therapies. This assessment has been carried out through physically administered [22,23,30,31,32,33,35,36], mailed [21,24], or web-based [26,27,33] questionnaires, usually author-developed [30,31,35], and based on available guidelines or fast stroke recognition codes or scales.

In recognition of the critical role played by ED in minimizing door-to-revascularization times and improving stroke outcomes, we conducted a study in the Republic of Cyprus. Patients with stroke, lacking contraindications for thrombolysis within the proper timeline, must receive early and accurate treatment. Leading organizations in stroke management guidelines emphasize creating stroke teams for optimal care [5,37]. As stroke teams are not established in Republic of Cyprus hospitals, and the direct treatment of stroke patients is a collective responsibility [38], our research aimed to assess stroke recognition and management knowledge among all healthcare professionals in the ED involved in stroke care. Recognizing the pivotal roles of both nurses and physicians in initial in-hospital stroke care within Greek Cypriot EDs, our study targeted both of these healthcare provider populations rather than specifying the role of each team member. This study marks the inaugural effort of its kind in Cyprus.

## 2. Materials and Methods

### 2.1. Design and Setting

This was a descriptive comparative study that invited all 12 EDs of public and private hospitals (seven and five hospitals, respectively) across the Republic of Cyprus to participate. All seven public (Hospital A, Hospital B, Hospital C, Hospital D, Hospital E, Hospital F, Hospital G) and four of the five private hospitals (Private Hospital A, Private Hospital B, Private Hospital C, Private Hospital D) provided consent. Strengthening the Reporting of Observational Studies in Epidemiology (STROBE) was followed [39].

### 2.2. Participants and Sample Size

The target population included HCPs, specifically nurses and physicians, who had worked at least one shift in an ED of a medical institution that granted consent. Participants needed to be able to read and write in Greek and provide written informed consent. No exclusion criteria were applied. A power analysis, conducted using G-Power statistical software (Version 3.1.9.4) [40], aimed to detect a small effect size (R^2^ = 0.06) related to the demographic characteristics of ED HCPs and their stroke knowledge. For this analysis, a linear regression model with 7 variables was employed. It was determined that to achieve 80% statistical power and maintain a 5% level of significance, a minimum sample size of 247 individuals was required. We implemented convenience sampling for enrollees, distributing a total of 399 questionnaires.

### 2.3. Data Collection and Instrument

Data collection occurred from November 2019 to April 2020, during which the research team distributed an anonymous self-administered paper questionnaire to participants in the EDs that took part in this study. Each ED was accessed at least three times during different shifts of the same day, and ED HCPs were invited to participate anonymously after a brief presentation of this study’s aim. The questionnaire, which began with an introductory cover letter providing a comprehensive explanation of this study and its objectives, comprised 37 questions covering the following areas:-Sociodemographic and employment characteristics (8 closed-ended questions): this section assessed factors as age, gender, profession, educational attainment, employment status, type of hospital (public or private), years of work experience, and previous education or training in stroke, along with the sources of such education.-Knowledge on stroke recognition and management (28 questions): this section included:
(a)General knowledge on stroke (GKS): four questions with responses: ‘right’, ‘wrong’, or ‘I don’t know’.(b)Knowledge on stroke recognition (KSR): two multiple-choice questions and six questions with responses: ‘right’, ‘wrong’, or ‘I don’t know’.(c)Knowledge on stroke management (KSM): two multiple-choice questions and 14 questions with responses: ‘right’, ‘wrong’, or ‘I don’t know’.-Self-assessment on knowledge of stroke recognition and management: one question with responses: ‘poorly’, ‘well’, ‘very well’, and ‘expert’.

Participants were not afforded any preparation time; they were required to complete the questionnaires under the supervision of a member of the research team within a maximum of 15 min. The 15 min time allocation for completing the 37-item questionnaire was determined through a thorough pilot test during the questionnaire development phase (refer to Section 2.4). Subsequently, the completed questionnaires were placed in anonymous envelopes, and the signed consent documents were stored separately.

### 2.4. Questionnaire Development

A pertinent questionnaire in the Greek language had not been previously created. Additionally, the existing literature did not adequately incorporate the latest stroke guidelines from the 2018 American Heart Association (AHA) and American Stroke Association (ASA) stroke guidelines [8], which were in effect during the study. In response to these gaps, we conducted a systematic process to develop a new tool for assessing knowledge of stroke recognition and management. This new instrument was designed to align with the 2018 AHA and ASA stroke guidelines [8], all adapted to the Greek language.

The questionnaire package was developed by study researchers through a staged procedure, according to the Delphi method [41], in which the following individuals were involved: the main researcher; a nurse with 4 years’ experience in a stroke care unit abroad; a stroke specialist/neurologist with 10 years’ experience in a dedicated stroke care center abroad; a professor of neurology of the Medical School of the University of Cyprus; an internal medicine specialist, with further specialization in stroke care at the Stroke Unit of the Royal Infirmary of Edinburgh; and a nurse with 10 years’ experience in EDs. During the first stage, the main researcher proposed 35 knowledge questions to the committee, based on the previously mentioned stroke guidelines, and 10 personal data ones. Three stages followed, during which the members of the committee graded the questions on a 5-point Likert scale, and, under the agreement that questions accumulatively scoring below 80% would be excluded, ended up with 28 knowledge and 8 personal data questions. These questions were then categorized into general inquiries (4 questions), questions related to stroke recognition (8 questions), and questions concerning stroke management and in-hospital follow-up (16 questions) (refer to Appendix A). The survey was initially completed by a convenience sample of 15 nurses in the ED of Hospital D, who were not included in this study’s final sample, and it was then redistributed 15 days later for internal consistency assessment.

### 2.5. Ethics Approval

This study was initially approved by the National Committee of Bioethics of Cyprus (EEBK/EP/2019.01.74) followed by approval from the Ministry of Health of the Republic of Cyprus (Approval Number: 0514/2019). Each hospital administration was then approached and consent for enrollment of personnel from the respective ED was requested. All participants provided their informed written consent. Furthermore, this study’s cover letter explicitly stated that all data would be kept deidentified, accessible solely to the research team, and securely stored in password-protected files. We also emphasized the voluntary nature of participation in this study. Each questionnaire package was delivered in unmarked, sealed envelopes without any identifiable features. This study was conducted in accordance with the principles outlined in the Declaration of Helsinki. It is important to note that no interventions were conducted on patients during the course of this study.

### 2.6. Data Analysis

The internal consistency of the tool was calculated using the Kuder–Richardson and interrater reliability with kappa statistic [42]. Kappa values, representing the agreement between the answers provided by the 15 participants during the first and second time the questionnaire package was distributed, were interpreted as follows: 0.00–0.20: no agreement; 0.21–0.40: satisfactory; 0.41–0.60: moderate; 0.61–0.80: strong; and 0.81–1.00: almost perfect [43]. The knowledge level of study participants, that is, the sum of correct answers across the 28 questionnaire items, was presented as mean +/− standard deviation. The correlations between stroke knowledge levels and the characteristics of participants were investigated through independent samples *t*-tests and ANOVA, comparing different groups of participants, according to, e.g., previous training on stroke care or years of work experience. Variables that were found to be statistically significantly associated with stroke knowledge in univariate analysis were then entered into a multivariate, linear regression model to identify independent predictors of stroke knowledge. Statistical analysis was performed using the Statistical Package for Social Sciences (SPSS), version 25.

## 3. Results

### 3.1. Questionnaire Testing

Kuder–Richardson was found to be 0.71 for the complete 28-item stroke knowledge questionnaire, 0.66 for the 4 general knowledge items, 0.45 for the 8 items regarding stroke recognition, and 0.62 for the remaining items on stroke treatment and follow-up. Table 1 presents the 28-item stroke knowledge questionnaire, translated in English, and the respective kappa values, all of which demonstrated an agreement percentage of over 70%, denoting acceptable interrater variability.

### 3.2. Sociodemographic and Employment Characteristics of Study Participants

A total of 281 questionnaires were returned completed (response rate (RR): 70.4%). In particular, response rate among nurses was 74.1% (255 questionnaires returned out of 344 distributed), while among physicians it was 47.3% (26 out of 55). Participants were predominantly female (53%); more than half (55.2%) belonged to the age group of 30–39 years. Regarding educational attainment, almost two out of three (61.9%) were university graduates, while a master’s degree had been obtained by 35.3% of respondents. Of the total, 70% of respondents had work experience of over 4 years in an ED and the largest contribution to the study sample came from the four state hospitals HB, HC, HD, and HE: 10.7%, 18.5%, 26.3%, and 10% of total returned questionnaires, respectively. Private hospitals accounted for 15.7% of study participants.

As for prior education or training relevant to stroke care, more than half of ED HCPs (55.9%) stated that they had received such training in the past. Table 2 presents the sources of such education or training.

### 3.3. Stroke Knowledge Levels

As shown in Table 3, both nurses and physicians fared poorly on the 28-item stroke knowledge questionnaire, with equally low scores in the stroke recognition and management sections compared to the general knowledge section. The distribution of scores followed a normal distribution, with the majority of nurses scoring between 10 and 20 and physicians between 15 and 20. Figure 1 presents the ranking of questionnaire items based on the percentage of correct answers. As expected, the majority of participants correctly answered the general knowledge questions (Questions 1–4). When considering physicians in isolation, the most frequently correct answer was associated with Question 7 (stroke mimics—hypoglycemia), with an impressive 96.2% accuracy. Conversely, the four items with the lowest scores (Questions 16, 26, 25, and 14, where correct answers were provided by only 7.1%, 13.2%, 16%, and 17.8% of participants, respectively) all pertained to thrombolysis. These questions covered topics such as the transfer of thrombolyzed patients to the intensive care unit (ICU), the acceptable body temperature to commence thrombolysis, the completion of blood workup and other necessary tests before the administration of thrombolysis, and the recommended thrombolysis time window.

Participant self-assessment on knowledge of stroke recognition and management revealed that 58.4% felt they performed ‘well’, 21% ‘very well’, and 20.6% ‘poorly’.

### 3.4. Univariate and Multivariate Predictors of Stroke Knowledge

The observed difference in stroke knowledge favoring ED physicians over nurses was statistically significant in both the total knowledge score (*p* < 0.001) and stroke recognition knowledge (*p* = 0.001). Prior participation in stroke education/training, reported by 137 nurses and 20 physicians, correlated with a statistically significantly higher stroke knowledge score, averaging 1.45 units. Univariate analysis revealed that other sociodemographic and employment characteristics such as gender, age, or hospital type (public vs. private) had no impact on performance in the knowledge questionnaire.

With regard to work experience, participants were divided into six groups, with <1, 1–5, 6–10, 11–20, 21–30, and >30 years of work experience in EDs. This variable, together with the aforementioned ones in univariate analysis, were entered into a multivariate linear regression model, the results of which are demonstrated in Appendix A. In particular, a statistically significant effect of the participants’ work experience on stroke knowledge was found, with all groups of work experience being superior to the <1 year of work experience group, and with increasing beta values as age increases (refer to Appendix A). Moreover, significant differences among participating hospitals were evident, with the HC exhibiting the highest scores, although no difference was, again, found collectively among public versus private hospitals. Surprisingly, educational attainment was not found to have a significant influence on stroke knowledge, except for a slight tendency towards statistical significance observed among those who had obtained a Master’s degree. The superiority of physicians’ knowledge versus nurses displayed in univariate analysis retained its statistical significance within the multivariate model, driven here mainly by a difference in stroke recognition/diagnosis. Finally, having prior education or training on stroke was once again found to be a statistically significant and independent predictor of higher levels of knowledge in general stroke knowledge, stroke recognition, and stroke management.

## 4. Discussion

The present survey aimed to assess stroke recognition and management knowledge among healthcare professionals in the ED for the first time in Cyprus. We evaluated the general knowledge of frontline health professionals caring for stroke patients. Our study did not aim to delve into specific individual roles or highlight the knowledge each healthcare professional should possess. We treated them as a collective group, and our discussion did not elaborate on separate responsibilities, with no prioritization of questions. We demonstrated relatively low levels of stroke knowledge among HCPs working in EDs throughout the Republic of Cyprus, using a newly developed and validated questionnaire. Multivariate analysis revealed that higher levels of comprehensive stroke knowledge were significantly associated with extensive experience, being a physician rather than a nurse, and prior education or training in stroke management.

The level of stroke knowledge among HCPs in the present study was found to be 46%, which is proportionally rated among the lowest in the available literature. Harper et al. demonstrated, in one of the first studies in the field, a mean score of 53% for stroke knowledge among 20 U.S. nurses who had completed a short, 10-item questionnaire [32]. In another U.S. study, 63 nurses and paramedics achieved an average score of 58% on a 10-item, evidence-based, multiple choice knowledge quiz [33]. A study in Brazil involved testing 20 nurses on the recognition of stroke signs and symptoms, resulting in an average score of 68.5% [29]. Emergency HCPs in Saudi Arabia [13] and India [30] achieved even higher scores, namely, 64% and 68.8%, respectively. A recent nationwide study from Malaysia, conducted with an online questionnaire among HCPs, found that 76% of respondents demonstrated a solid understanding of stroke [44].

Comparing stroke knowledge levels across various studies can be challenging due to several factors: (i) the inclusion of diverse healthcare professionals, such as nurses, paramedics, physicians, and even medical students [36] with varying training backgrounds; (ii) the wide range of stroke care settings, including dedicated stroke care units, emergency departments, or prehospital emergency services; (iii) the lack of universal tools for quantifying stroke knowledge, with most studies using author-developed tools specific to their research; and (iv) study populations originating from different countries, resulting in substantial differences in stroke care organization and significant variations in the implementation of guideline-based therapies. Despite these shortcomings, a universal conclusion can be drawn: there is an overall suboptimal level of stroke knowledge among healthcare professionals engaged in stroke care.

Insufficient knowledge has been identified as a barrier to providing evidence-based stroke care [45]. Demonstrably, educational programs for HCPs have been shown to enhance their knowledge and proficiency in stroke care [46]. While these educational initiatives are crucial, a systematic review investigating the impact of stroke education and training for HCPs involved in stroke care reveals that the precise effect on patient outcomes remains unclear. Limited survey results indicate the need for more comprehensive research to understand the direct impact on patient wellbeing. Nevertheless, these aspects are acknowledged as crucial to maintaining a high standard of care for stroke patients. Notably, they improve the ability to recognize stroke, thereby increasing the number of strokes identified by HCPs [47].

Concerning stroke recognition codes, only 24.9% of our study respondents identified the NIHSS as the proposed tool for assessing stroke severity (Question 5), indicating a likely unfamiliarity with the scale. This aligns with Lamba et al.’s study, where 62% of U.S. ED workers reported being unfamiliar with the NIHSS [23], and with a study in a rural hospital in Brazil, where only 31.25% of ED nurses were familiar with the NIHSS [29]. In contrast, Reynolds et al. found elevated levels of knowledge concerning the NIHSS (88.6%) among 88 highly specialized nurses in neurocritical care in a U.S. university hospital [35]. Given these findings, there is a crucial need for disseminating material on the NIHSS, coupled with dedicated, focused, and repeated training in its completion across diverse clinical scenarios within Greek Cypriot EDs.

Early recognition is the cornerstone of stroke therapy, and studies have shown how early recognition and therapy affect optimal patient outcomes [48,49]. Moreover, the use of NIHSS as an accurate and early diagnostic tool has demonstrated its efficiency when used by emergency medical services, providing a common language between healthcare professionals [50].

Examining specific aspects of stroke management knowledge, our study revealed that only 17.8% of participants were aware of the guideline-proposed time window for thrombolysis in stroke patients (Question 14). This percentage is notably lower than those reported in similar studies. For instance, in a U.S. academic tertiary hospital study involving 58 emergency department healthcare providers, 56% of respondents were familiar with the 3 h thrombolysis time window [23]. In contrast, in a stroke referral center in Kenya, 53.8% of respondents were aware of it, although two-thirds of the participants were medical students [36]. Similar results were observed in a Chinese study, where 54% of community physicians were aware of the thrombolysis time window [51]. Greek Cypriot ED HCPs demonstrated extremely low awareness of the current guideline for the 3 h thrombolysis time window, and this lack of knowledge extended to other thrombolysis-related questionnaire items, specifically Questions 16, 21, 25, and 26 (refer to Figure 1 and Table 1). Albart et al. also reported lower knowledge results regarding thrombolysis, despite high overall knowledge scores [44], highlighting the urgent need for targeted training and education on thrombolysis for HCPs involved in the care of stroke patients.

Suboptimal knowledge concerning the therapeutic window for thrombolysis in our study is alarming, as the neurological outcome is significantly affected by the early initiation of the intervention [52].

A significant and rather linear association was found between years of work experience and performance in the stroke knowledge test, establishing it as an independent predictor in multivariate analysis across all groups of work experience, particularly compared to those with <1 year experience. This finding aligns with previous research. Specifically, Harper et al. demonstrated a positive correlation between more years of experience in emergency nursing and higher knowledge scores [32]. Clinical experience among sub-Saharan nurses emerged as the most significant predictor of specific knowledge or skills, such as selecting the appropriate IV fluid for stroke patients or adhering to the thrombolysis time window, as indicated by Lin et al. [36]. However, this observation has not been consistent across all studies. A Polish study revealed that paramedics with less than 11 years of experience exhibited greater proficiency compared to their more experienced colleagues. The finding was attributed to recent training and adherence to up-to-date guidelines [53]. Adelman et al., in a sizable sample of 875 nurses from a single U.S. academic center, found no association between clinical experience (categorized as <1, 1–3, 4–10, and ≥11 years of employment) and adequate knowledge on stroke warning signs [34]. It is worth noting that this study primarily focused on early recognition through warning signs, omitting other aspects of stroke awareness such as thrombolysis issues or patient management thereafter. A broader stroke knowledge base could possibly have discriminated an experienced from an inexperienced health professional involved in stroke care.

With regard to the contribution of previous stroke education or training on higher stroke knowledge levels, a 2009 study demonstrated that reading relevant literature on stroke and participating in continued medical education (CME) activities were associated with higher stroke knowledge by up to 45% and 15%, respectively, in U.S. nurses [32]. However, another U.S. study found that studying relevant material in the preceding year did not provide benefit to nurses, unlike participation in CME and being a certified ED nurse [33]. In our study, educational attainment did not affect stroke knowledge levels but prior education or training did. However, we did not investigate how different sources of prior exposure to stroke knowledge (self-study, congress workshop, etc.) influenced achieved scores. From an organizational perspective, it would be useful to know which interventions aid the most in building confidence in stroke care. This knowledge could guide hospitals in providing more targeted educational interventions to their ED staff.

In undergraduate medical students, the observation of stroke cases appears to be limited, underscoring the need for better demonstrations [54]. Opting for a stroke-related fellowship could enhance the education of medical students [55]. Regarding nursing students, evidence on the level of education is limited, but it seems that there are differential methods for stroke training across universities [56]. While the educational challenges among medical and nursing students are evident, our study delved into the practical implications for physicians and nurses. With respect to the observed differences in our study among nurses and physicians, no study, to the best of our knowledge, has investigated this matter to date. Albart et al. compared knowledge across various physician categories and other HCPs, including nurses, but did not provide a clear indication of the nurses’ scores [44]. The higher stroke knowledge among physicians compared to nurses noted in our study should be interpreted with caution and not generalized. Notably, a significantly lower response rate among Greek Cypriot ED physicians (47.3%) compared to nurses (74.1%) was observed. This discrepancy may be related to a lack of willingness by a significant number of physicians with lower levels of stroke knowledge to participate, potentially influencing the reported knowledge differential.

Regarding response rates, our study faces challenges in direct comparison with those reported in the existing literature. The diverse types of healthcare professionals included in studied populations and the utilization of various enrollment methods (such as online surveys, face-to-face recruitment, questionnaires sent by regular or electronic mail, etc.) contribute to this complexity. For instance, a study that included 875 inpatient and ED nurses from a large academic hospital in the U.S., recruited via an online survey, displayed an overall response rate of 84% [34]. Two studies focusing solely on emergency physicians, one in Saudi Arabia [27] and the other in Australasia [26], both web-based, yielded response rates of only 27% and 13%, respectively. However, comparing the response rate of physicians in our study (47.3%) demands careful scrutiny due to the utilization of face-to-face recruitment, which is expected to be more effective in recruiting participants. The differing recruitment methods make a direct comparison difficult. Lastly, in the 1999 study by Thomas et al., invitation to participate was sent by regular mail to nurses in North East England, resulting in a response rate of 80% [31].

Our study was conducted from November 2019 to April 2020, largely pre-pandemic, as the World Health Organization declared the COVID-19 pandemic on 11 March 2020. Therefore, the pandemic did not affect the data collection or the knowledge of health professionals. Early pandemic visits in EDs were shown to be lower for reasons other than virus-related diseases, as compared to the pre-pandemic period [57]. Moreover, virus incidence was still low at that time, and mainly affected the ICU occupancy in Cyprus rather than EDs [58].

## 5. Limitations

While our study provides valuable insights into knowledge on stroke recognition and management among emergency department healthcare professionals in the Republic of Cyprus, it is imperative to acknowledge various limitations that warrant consideration.

Selection bias may not be excluded due to the voluntary nature of participation. This affects the ability to generalize the study findings, especially in the case of physicians. Nevertheless, it should be noted that the recruitment process included all health professionals in emergency departments across public and private hospitals in Cyprus (with only one private hospital not agreeing to participate) with a response rate of around 75%. However, there was a notable difference in the response rates between nurses (74.1%) and physicians (47.3%), which may impact the external validity. This underrepresentation of physicians in the sample may impact the generalizability of our findings to the broader physician population.

It should also be noted that the knowledge deficit on stoke recognition and management care identified in the study might be an underestimation, if we assume that healthcare professionals who opted out from participating may be less interested in the topic and/or uncertain about their stroke knowledge. Therefore, it is prudent to acknowledge that actual knowledge levels may be even lower, necessitating a greater effort to take action.

Another limitation, also affecting external validity, is the underrepresentation of private hospitals in the final sample, accounting for only 15.7% of respondents. However, the majority of stroke cases in the Republic of Cyprus are directed by the emergency medical services to the EDs of public hospitals, where treatment is administered. This renders the relatively low representation of private hospitals reasonable. While the refusal of an entire private hospital to participate in this study may have resulted in a slight overestimation of overall stroke knowledge levels, its impact on the generalizability of this study is expected to be minimal.

Finally, the questionnaire used to assess stroke knowledge was author-developed for this specific study and has not been tested previously. Nevertheless, it was developed following best practices in survey design and stroke guidelines. The questionnaire also demonstrated acceptable internal consistency, with a Kuder–Richardson coefficient of 0.71, nearly identical to the one found in the questionnaires developed by Thomas et al. (0.7). It is worth noting that both questionnaires were created by a multidisciplinary team of experts and were based on current guidelines [31].

## 6. Conclusions

With the utilization of a newly developed questionnaire based on current knowledge regarding the recognition and management of patients with stroke, our study revealed significant gaps in the stroke knowledge levels of ED nurses and physicians in the Republic of Cyprus. Previous successful efforts to enhance stroke knowledge among health professionals through educational or training interventions underscore the potential for improvement. While experience can provide advantages over time, certain aspects of stroke knowledge, such as thrombolysis, require urgent attention.

To address these gaps, we recommend the implementation of dedicated stroke guideline courses, possibly offered by the Ministry of Health. Workshops providing hands-on training on practical aspects, including the implementation of scales like the NIHSS in various clinical scenarios, are essential. Additionally, we propose that hospital and prehospital staff involved in stroke care undergo yearly examinations with formal stroke knowledge tests. These interventions should be ongoing and repeated at regular intervals, considering the evolving nature of knowledge and practice guidelines, as well as the turnover of ED staff. It is crucial to focus on the entire ED staff without discrimination, as both nurses and physicians share a uniformly low stroke knowledge base within the Greek Cypriot EDs, along with the common responsibility of optimally managing stroke patients.

## Figures and Tables

**Figure 1 healthcare-12-00077-f001:**
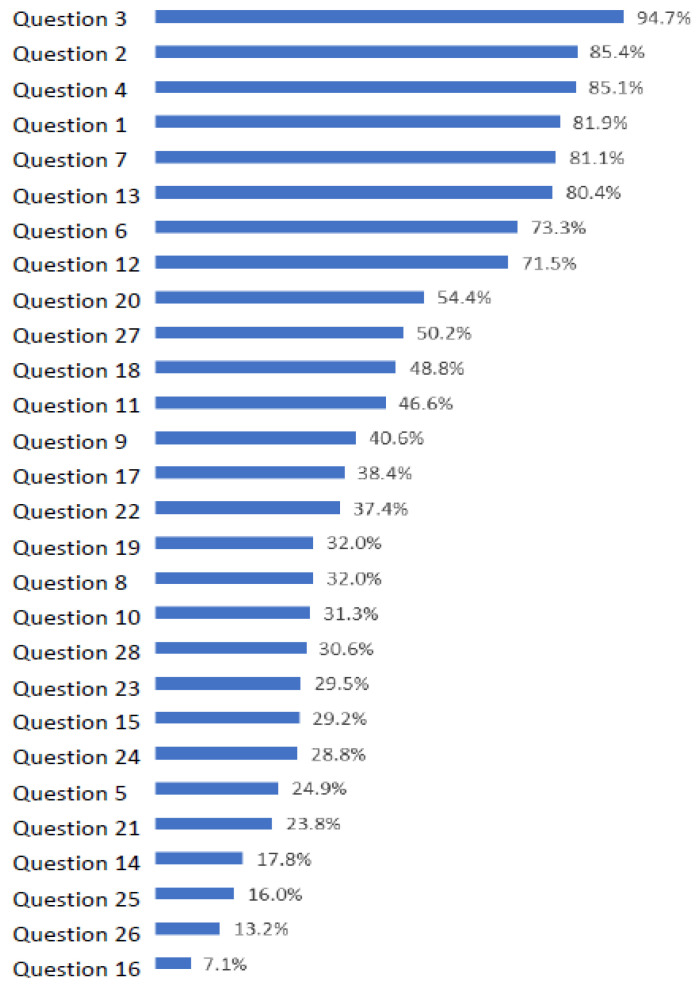
Ranking of questions of stroke knowledge questionnaire according to percentage of participant that answered correctly (N = 281).

**Table 1 healthcare-12-00077-t001:** Twenty-eight-item, author-developed questionnaire on stroke knowledge and respective kappa values and agreement percentages (run-in sample, N = 15).

Question	Kappa	Agreement Percentage
Every patient presenting with stroke should be treated as a medical emergency, whether eligible for thrombolysis or not (GKS)	0.73	87%
2. Stroke represents a life-threatening situation (GKS)	0.86	93%
3. EDs play a vital role in the early recognition of stroke and the timely commencement of treatment (GKS)	1.00	100%
4. EMS personnel should inform ED about the transfer of a patient with a probable stroke (GKS)	0.33	73%
5. NIHSS is the proposed scale for stroke severity assessment (KSR)	0.55	80%
6. A patient presenting to the ED with stroke is most likely to experience which of the following symptoms? (i) tremor, dizziness, vomiting; (ii) altered level of consciousness, tachypnea, cyanosis; (iii) unilateral arm or leg weakness or face drooping or difficulty speaking (KSR)	0.47	80%
7. Which of the following can mimic stroke? (i) hypoglycemia, (ii) hyperkalemia, (iii) heat stroke, (iv) pulmonary embolism (KSR)	0.59	87%
8. Brain MRI is the recommended diagnostic modality to differentiate between ischemic and hemorrhagic stroke (KSR)	0.48	73%
9. In a patient with suspected stroke, clinical examination and history taking can securely differentiate between ischemic and hemorrhagic stroke (KSR)	0.21	67%
10. 60 min is the maximum time allowed from the arrival of the patient at the ED and the commencement of diagnostic examinations (KSR)	0.71	87%
11. A patient with suspected stroke, regardless of severity and neurological deficit, must be placed in bed, supine (KSR)	1.00	100%
12. In patients with suspected stroke, blood pressure should be measured on both arms and a finger stick glucose test performed (KSR)	0.59	80%
13. Hypotension and hypovolemia should be treated before starting thrombolysis (KSM)	1.00	100%
14. Thrombolysis must be administered to eligible stroke patients within a time window of 3 h (KSM)	0.57	80%
15. Patients aged over 80 years should be excluded from thrombolysis (KSM)	0.47	80%
16. In a stroke patient about to receive thrombolysis, the maximum acceptable body temperature is (i) 37 °C, (ii) 37.5 °C, (iii) 38 °C, (iv) 38.5 °C (KSM)	0.25	73%
17. The lowest acceptable blood glucose level for a patient about to receive thrombolysis is 60 mg/dL (KSM)	0.86	93%
18. If the patient who is about to receive thrombolysis demonstrates an oxygen saturation of <94%, we administer oxygen via nasal cannula and proceed with thrombolysis as planned (KSM)	0.66	87%
19. What is the recommended dose of rt-PA in a stroke patient? (i) 0.9 mg/kg, (ii) 90 mg/kg, (iii) 0.6 mg/kg; (iv) 1.3 mg/kg (KSM)	0.86	93%
20. What is the maximum acceptable blood pressure before administering thrombolysis? (i) 200/115 mmHg; (ii) 230/115 mmHg; (iii) 215/120 mmHg; (iv) 185/110 mmHg (KSM)	0.42	87%
21. During the administration of thrombolysis, blood pressure must be measured every 30 min (KSM)	0.66	87%
22. Administration of aspirin is recommended within 24 to 48 h after thrombolysis (KSM)	0.81	93%
23. Thrombolysis can be administered to a stroke patient who is receiving a therapeutic dose of heparin (KSM)	0.42	87%
24. Maximum allowed dose of r-tPA is 40 mg (KSM)	0.84	93%
25. For thrombolysis to be administered, blood tests, chest X-ray, and electrocardiogram must all be completed (KSM)	0.81	93%
26. After thrombolysis treatment, the patient must be transferred to an ICU for 12 h (KSM)	1.00	100%
27. In a patient who has received thrombolysis, vital signs should be taken regularly only during the first 12 h receiving r-tPA (KSM)	1.00	100%
28. In a patient with a severe stroke and large-vessel occlusion, thus with an indication for thrombectomy, we immediately administer thrombolysis (if the patient is eligible for that) and thrombectomy follows (KSM)	0.86	93%

ED: emergency department; EMS: emergency medical services; GKS: general knowledge on stroke; ICU: intensive care unit; KSM: knowledge on stroke management; KSR: knowledge on stroke recognition; NIHSS: National Institutes of Health Stroke Scale; MRI: magnetic resonance imaging; r-tPA: recombinant tissue plasminogen activator.

**Table 2 healthcare-12-00077-t002:** Sources of prior education/training in stroke care among emergency department healthcare professionals (Ν = 157 *).

	Nurses	Physicians	Total
If you had any prior education or training relevant to stroke care, what was the source of such education or training?	Ν (%)	Ν (%)	Ν (%)
Self-guided study	3 (2.2)	2 (10)	5 (3.2)
Congress/educational workshop	49 (35.8)	8 (40)	57 (36.3)
Class (postgraduate level)	6 (4.4)	1 (5)	7 (4.5)
Class (undergraduate level)	38 (27.7)	8 (40)	46 (29.3)
Brief presentation on stroke care	37 (27)	0 (0)	37 (23.5)
Leaflet/other printed material	4 (2.9)	1 (5)	5 (3.2)
Total	137 (100)	20 (100)	157 (100)

* Only respondents who stated they had had relevant stroke education/training in the past were included in this table.

**Table 3 healthcare-12-00077-t003:** Performance of study participants in the 28-item stroke knowledge questionnaire (N = 281).

	Nurses	Physicians	Total
	Mean (SD)	Mean (SD)	Mean (SD)
Total score	12.6 (4.1)	15.7 (4)	12.9 (4.2)
General knowledge on stroke (4 items)	3.5 (0.9)	3.7 (0.9)	3.5 (0.9)
Knowledge on stroke recognition (8 items)	3.9 (1.6)	5 (1.5)	4 (1.6)
Knowledge on stroke management (16 items)	5.2 (2.7)	7 (2.4)	5.4 (2.7)

SD: Standard Deviation.

## Data Availability

The data presented in this study are available upon request from the corresponding author.

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
