# Peer review of "Knowledge on Stroke Recognition and Management among Emergency Department Healthcare Professionals in the Republic of Cyprus"

_healthcare, 2023, doi:10.3390/healthcare12010077_

Round 1

Reviewer 1 Report

Comments and Suggestions for Authors

Overall, I find this work to be of good quality, with a well-structured introduction, clear materials and methods section, a thorough description of questionnaire development, and a commendable data analysis presentation. However, there are a few points where the paper can be improved:

The introduction is detailed and provides a comprehensive background for the study. However, there is an ambiguity in sentence 66, which states, "among healthcare professionals working in the ED for the first time in Cyprus." It is unclear whether this refers to the study being conducted for the first time in Cyprus among ED professionals (supposedly intended meaning) or if it is about ED professionals working in Cyprus for the first time. This ambiguity should be clarified for better understanding.

The Materials and Methods section is well-explained and clear. It provides a good overview of the study design and data collection process.

The section on questionnaire development adequately describes the process, offering transparency regarding how the research instrument was designed and validated. 

The data analysis and reports section is presented with high quality and employs appropriate statistical methods. This aspect of the study is particularly commendable as it ensures the validity of the findings.

There is a typographical error in line 357, which states, "this could be related to a LACK of willingness..." 

Limitations section:

The first two lines in the Limitations section (lines 363 and 364) appear to be redundant and may be omitted or moved to the "Discussion" section.

The content between lines 375 and 384, currently found in the Limitations section, would be more appropriately placed in the "Discussion" section, as it compares the authors' findings to previously conducted studies.

The underrepresentation of physicians versus nurses (26 vs. 255) is mentioned in the paper but is not explicitly addressed in the Limitations section. 

In conclusion, this work is a well-executed study with a strong foundation in its introduction, methodology, and data analysis.

I recommend that the authors consider these suggestions to enhance the overall quality and clarity of their work.

Comments on the Quality of English Language

Good quality of English language, a few typos and minor mistakes only

Reviewer 2 Report

Comments and Suggestions for Authors

The paper by Rossis et al. investigates prehospital emergency care personnel familiarity with stroke management.

This is an important field, as personnel is expected to provide the patients with the choices that offer the most chances of survival and\or minimum disability impact; therefore, it is paramount that EBM and guideline guidances are followed.

On one hand, the paper has merit in terms of general structuring, is well presented and clear. Its results appear coherent with current literature.

On the other hand, there are some methodological points to be raised. Generally speaking, questionnaires are too often seen as a cheap and almost effortless way of obtaining information, and are often treated lightly by researchers. This is sub-opbtimal from a scientific standpoint, as it basically makes this method prone to avoidable errorsand they need extensive planning, time and effort in order to obtain meaningful results.

My main concern is that the questionnaire, despite being quite sound from a health operator perspective, looks amateurish from a developement perspective. To beeter explain what I mean, just compare some freely available papers on questionnaire developement, such as these (https://www.ncbi.nlm.nih.gov/pmc/articles/PMC8088187/ or https://www.ncbi.nlm.nih.gov/pmc/articles/PMC420179/), with the amount of information provided here by the Authors.

I also have some doubts in the general alnalysis approach, wich I will sum up in the following points.

1) The Authors used a purpose-built questionnaire, consisting of 37 item, to be answered in 15 min. How was this timing chosen?

2) Has any validation been performed? there is no mention of internal nor external validation in the paper.

3) The Authors mention to have measured inter-rater agreement using Cohen's k. This is correct when comparing answers whose correctness is not defined a priori. Its interpretation in current framework is quite controversial however, as many quesitons are related to well-known guidelines, for wich a correct answer exists, and all other are incorrect. If 100% of the interviewed answer the same wrong quesiton, k will be 1 and agreement will be perfect, but it will still mean that 100% answered wrong. Authors should either rethink this approach, or provide the reader guidance with the interpretation of the results.

4) Likewise, usage of Cronbach's alpha is likely suboptimal for the Authors' purposes, as it measures reliability rather than actual agreement. Cfr. https://doi.org/10.1037/0021-9010.78.1.98. Also, among its assumptions, it requires the data to be normally distributed and linear (i.e. non nominal). This is clearly not the case, as seen in the following point.

5) Several questionnaire items appear to have been considered as continuous variables in the analysis (t-test and linear regression are tipycal of such cases). However, each point of each item is not to be purely treated as a mere number; In most cases, they are Ordinal variables. Some question even provides just categorical answers (ex. 6 or 7). This means that the knowledge level measured as a mere sum of correct answer is, at best, a poor estimation, and some other strategy may be used to support this measurement; for example, z-scores for each item before summing up could be a possible workaround. This howewer would not carry the notion of "distance" from a correct answer, but only the ratio of correct ones for each person, in relation to the whole sample; therefore should again be treaded on carefully. Discussion should again mention this.

6) Sampling strategy did account for different turns, wich is good, and for different facilities (again good). However, in subsequent analysis, the notion of different facilities got lost. This creates a possible clustering bias, therefore any analysis should be performed using not a simple omdel, but a multilevel (nested) one (responder<facility).

7) On top of that, answers should be weighted in all analysis, according to sample representativeness. Weights should be calculated differently for each facility and professional figure, as (n° of actual responder in the sample)/(n° total potential responders in the population). Cfr. https://link.springer.com/chapter/10.1007/978-3-319-54395-6_21

Reviewer 3 Report

Comments and Suggestions for Authors

Thankyou for the opportunity to review your manuscript on a novel survey to access doctor & nurse understanding of ischaemic stroke. 

abstract. some simple errors eg "15,7" should be "15.7", SD: 4.2 and SD:4.1 (spacing is inconsistent). momre details around the questionirre are needed - were some questions more important to get right than others?

Introduction.  succint. it would be good to understand what sort of stroke detection education exists within the literature. It would be good for international readers to understand what the 'standard stroke workup' is in the ED and who has decision making capacity to perform thrombolysis (eg is it the ED physician who is being surveyed, or is it the stroke team who is not being surveyed? this would create bias in the results). 

methods. 2.4 instrument should be a supplementary with key parts of the survey being included in the manuscript only. it was interesting to see the questionairre being performed peri-pandemic!

results. well presented. Table 1. is exhausting/difficult to read

Discussion. a thourough comparison with the literature and comparing with findings. i would like to see some discussion around the impact and patient outcomes that may reflect survey results (if any). Also determining the individual responsibilities and what they should know as opposed to a survey that seeks to be a 'one size fits all'. Eg need to have a sharp acumen at triage to identify stroke, ed doctors need to understand the time critical natures and options for intervention, stroke doctors need the deepest understanding of stroke management. It would be good to discuss COVID-19 effects on survey repsonses

Conclusion. The conclusion should not introduce new concept. please move the idea of education into the discussion. You also include a pre/post study in the conclusion - this should be moved or removed as there is no reference to it. 

Comments on the Quality of English Language

incorrect terms used throughout, please check eg "3.2 Characteristics of the srudy participants' 

Author Response

please the attachment

Round 2

Reviewer 2 Report

Comments and Suggestions for Authors

All my previous comments have been assessed. I have no further remarks.

Reviewer 3 Report

Comments and Suggestions for Authors

Thankyou for taking on board the comments from the first revision. The tone and pitch of the manuscript is now sufficient for publication.